# Improving Mobile Robot Maneuver Performance Using Fractional-Order Controller

**DOI:** 10.3390/s23063191

**Published:** 2023-03-16

**Authors:** Daniel Acosta, Bibiana Fariña, Jonay Toledo, Leopoldo Acosta

**Affiliations:** Computer Science and System Department, Universidad de La Laguna, 38200 Canary Island, Spain; alu0100908480@ull.edu.es (D.A.); bfarinaj@ull.edu.es (B.F.); jttoledo@ull.edu.es (J.T.)

**Keywords:** fractional control, autonomous vehicle, robotics

## Abstract

In this paper, the low-level velocity controller of an autonomous vehicle is studied. The performance of the traditional controller used in this kind of system, a PID, is analyzed. This kind of controller cannot follow ramp references without error, so when the reference implies a change in the speed, the vehicle cannot follow the proposed reference, and there is a significant difference between the actual and desired vehicle behaviors. A fractional controller is proposed which changes the ordinary dynamics allowing faster responses for small times, at the cost of slower responses for large times. The idea is to take advantage of this fact to follow fast setpoint changes with a smaller error than that obtained with a classic non-fractional PI controller. Using this controller, the vehicle can follow variable speed references with zero stationary error, significantly reducing the difference between reference and actual vehicle behavior. The paper presents the fractional controller, studies its stability in function of the fractional parameters, designs the controller, and tests its stability. The designed controller is tested on a real prototype, and its behavior is compared to a standard PID controller. The designed fractional PID controller overcomes the results of the standard PID controller.

## 1. Introduction

Mobile robotics is a very active research area. This includes the design and implementation of autonomous robots. These robots are capable of making intelligent decisions based on localization, path planning, obstacle detection and avoidance, and environment analysis modules. One of the key parameters for the success of a mobile robot is robot control. The robot must obey the decisions made by higher control layers in the most precise way. Any variation between the maneuver received and the actual maneuver executed can result in final application failure and more complicated high-level control.

Our research group has been working for some time with a low-cost electric golf cart [1]. The objective is to turn a standard golf cart into an autonomous vehicle so that some mechanical and electric modifications were made on it. The drive that generates traction is a direct current motor and a drive by wire steering system that coexists with manual steering is included. The prototype includes an on board computer, sensors and software that turn it into an autonomous robot capable of transporting two passengers in non-structured environments. The vehicle localize itself [2,3], makes navigation decisions [4,5], detects obstacles [6], avoids them [7,8], and plans the best route in real time [9]. The robot applies this plan using a steering and velocity control, and the quality of the control limits the final performance of the vehicle. This turns this vehicle into a good framework to test different self-driven vehicle strategies.

The sensor set includes two encoders attached to each rear wheel to obtain odometric information, an IMU, a centimeter GPS, three Lidars, and a stereo vision system. The software is developed on Robotic Operative System (ROS). The software is structured in layers, from the low level where there are sensors and actuators to the high level where there are planning layers able to make intelligent decisions based on the environment. Figure 1 shows the prototype, a fully electric two-seat golf cart.

The measured speed for the control is obtained from the odometric system of the prototype. The odometric system is based on encoders coupled to rear wheels, as shown in Figure 2. Each encoder provides 1024 pulses per revolution and each revolution of the wheel generates a revolution of the encoder (1:1 coupling). Wheel rotation is transferred to the encoders through a flexible mechanical transmission system that goes from the center of each wheel to the encoder placed on the side of the vehicle (see Figure 2). Encoder output is connected to an ad hoc electronics that samples the encoders signal every 0.5 ms. The electronics is designed to measure and integrate the encoder signals and the output is transmitted to the on-board computer at every integration period of 20 ms. The integration is made in the microcontroller installed in the ad hoc electronics, based on Euler integration, and collecting encoder increments for the integration time.

This paper focuses on the lower software level of the vehicle, the motor traction control when a trajectory is being tracked. The traditional way of approaching the control of a system is using a PID controller [10]. It is a convenient and easy way to apply a solution for controlling any system, but usually the performance of the controller system is not the best. To improve the control quality of the final system, some alternatives are available in the literature. Specifically, the classic PI controller used in the first designs was replaced by a new fractional control in order to achieve better maneuverability under certain conditions.

In particular, the control engineering benefited from the advantages of adding fractional operators to controllers. Incorporating integral and derivative fractional parts into a controller makes it possible to have two additional parameters to tune compared to the integer versions. These two parameters are the corresponding fractional orders. The objective is precisely to take advantage of the fractional controllers to obtain a better performance in the maneuverability of the prototype.

In this paper, a new fractional controller is used as the speed control for the autonomous vehicle. This controller allows it to follow the applied commands in a more precise way. Specifically, if the command sent by the high-level control is a speed increase or reduction, a traditional PID is not able to follow the trajectory without a stationary error, so the actual speed is different from the desired speed, as shown in Section 5. The fractional controller proposed is able to follow these variable speed commands with zero stationary error, and the error between reference and command is reduced as Section 7 results shows. This advantage allows a more precise and accurate movement of the autonomous cart.

Figure 3 shows the implementation of the low-level controller that is made in the autonomous vehicle, where a standard PI controller is substituted by a fractional PI.

## 2. Previous Work

Fractional calculus studies the generalization of integer-order derivatives and integrals to a fractional-order derivatives and integrals. This means that traditional calculus use integer indices in its derivatives and integrals, however, the fractional calculus allows to use fractional derivatives and integrals describing a more complex function. The fractional calculus may be considered an old and yet novel topic. It is an old topic because, starting from some speculations of G.W. Leibniz in 1695 and L. Euler in 1730, it has been progressively developed up to now. However, it may also be considered a novel topic because its applications began in recent decades. A complete description of the fractional mathematics can be found in [11]. In [12], a survey with the advances in fractional calculus since the 1970s is shown, which includes numerical applications to implement the actual fractional systems that can work in real time. In [13], a survey of many applications of fractional calculus, examples, and possible implementations are presented. It also contains a separate chapter of fractional order control systems, which opens new perspectives in control theory.

Fractional models have been applied to different problems to characterize the dynamics of processes with complex behaviors, as in [14], where fractional kinetic equations of the diffusion are presented as a useful approach for the description of transport dynamics in complex systems which are governed by anomalous diffusion and non-exponential relaxation patterns. Methods to find the solution are introduced, and for some special cases, exact solutions are calculated. This report demonstrates that fractional equations have come of age as a complementary tool in the description of anomalous transport processes.

In [15], fractional calculus is applied to the control of a revolute planar robotic manipulator. The fractional derivatives required by the control can be obtained by adopting numerical real-time signal processing. Numerical experiments illustrated the feasibility and effectiveness of the approach. Ref. [16] presents the possibilities of fractional calculus applied to system identification and control engineering, but also into sensing and filtering domains. The fractional-order electronic component has led to the possibility of analog filtering techniques from a practical perspective, enlarging the horizon to a wider frequency range, with increased robustness to component variation, stability, and noise reduction. Fractional-order digital filters have developed to provide an alternative solution to higher-order integer-order filters, with increased design flexibility and better performance.

The control of autonomous vehicles includes multiple steps, including route planning, behavioral decision-making, motion planning, and vehicle control [17]. The last step, vehicle control is usually made with a standard PID controller. A survey of the different strategies applied in the low-level control of the vehicles can be found in chapter 2 of reference [18], where the authors distinguish between model-based and model-free controllers. Model-based controllers are more complicated to implement, and when the system changes, as for example for battery discharge, its performance is reduced. However, model-free controllers are more difficult to adjust, but more robust to changes in the model. The fractional controller presented in this paper can be classified as model-free, but with better performance than standard PID controllers. In [19], the longitudinal control task is addressed by implementing adaptive PID control using two different approaches: genetic algorithms (GA-PID) and then neural networks (NN-PID), respectively, adapting the controller to the non-linearities and the change in system characteristics. In [20], a control schema to manage low-level vehicle actuators (steering throttle and brake) based on fuzzy logic, an artificial intelligence technique that is able to mimic human procedural behavior is presented, in this case, when performing the driving task.

In this paper, a new approach to controlling the speed of an autonomous robot is presented, where the fractional-order controller is used to improve the performance in reference tracking. The advantages of this kind of controller include the fact that it allows to obtain a better performance in robot tracking the following sections will show.

## 3. **Fractional Integral and Derivative**

Given a real function dependent on time f(t), its fractional integral $\left(I_{0^{+}}^{\alpha }f\right)$$\left(t\right)$ of order α is defined as Equation (Equation 1).
(1)$$\left(I_{0^{+}}^{\alpha }f\right)\left(t\right)≜\frac{1}{\Gamma \left(\alpha \right)}\int_{0}^{t}\frac{f\left(\tau \right)}{{\left(t-\tau \right)}^{1-\alpha }}d\tau$$
where α is the real positive integration order and $\Gamma \left(\alpha \right)$ is the Gamma function. The Laplace transform of this integral equation can be defined as Equation (Equation 2).
(2)$$L\left\{I^{\alpha }f\left(t\right)\right\}=\int_{0}^{\infty }e^{-st}\left(\frac{1}{\Gamma \left(\alpha \right)}\int_{0}^{t}\frac{f\left(\tau \right)}{{\left(t-\tau \right)}^{1-\alpha }}d\tau \right)dt=\frac{1}{s^{\alpha }}F\left(s\right)$$
with $I_{0^{+}}^{\alpha }\equiv I^{\alpha }$ and zero initial conditions.

The definition of the fractional integral is unique. However, for the definition of the fractional derivative, there are various proposals.

The Lagrange’s rule for differential operators is used to define the Riemann–Liouville fractional derivative ${}^{RL}D_{0^{+}}^{\beta }f\left(t\right)$ of order β for a function f(t). Given n∈N such that n−1<β≤n, the Riemann–Liouville derivative is obtained, computing the n-th order derivative over the integral of order (*n*−β) which is defined in Equation (Equation 3).
(3)$${}^{RL}D_{0^{+}}^{\beta }f\left(t\right)≜D^{n}\left(I_{0^{+}}^{n-\beta }f\right)\left(t\right)$$
where $D\equiv \frac{d}{\mathrm{dt}}$ and ${}^{RL}D_{0^{+}}^{\beta }\equiv D^{\beta }$ is used.

In a very similar way to the previous definition, changing the order of the derivative and the integral, it is possible to define the Caputo fractional derivative ${}^{c}D_{0^{+}}^{\beta }f\left(t\right)$ of order β in Equation (Equation 4).
(4)$${}^{c}D_{0^{+}}^{\beta }f\left(t\right)≜\left(I_{0^{+}}^{n-\beta }\left(D^{n}f\right)\right)\left(t\right)$$

The advantage of the Caputo derivative over the Riemann Liouville derivative, Equation (Equation 4), is that it is not necessary to define the fractional-order initial conditions when solving differential equations.

Another alternative definition for the fractional derivative is that of Grünwald–Letnikov ${}^{GL}D_{0^{+}}^{\beta }f\left(t\right)$ (Equation (Equation 5)).
(5)GLD0+βft=limh→01hβ∑j=0∞−1jβjft+β−jh
where βj is defined in Equation (Equation 6).
(6)βj=Γβ+1Γj+1Γβ−j+1

It can be shown that the above definitions of the fractional derivative are equivalent for a wide class of functions [13].

The Laplace transform of the fractional derivative $D^{\beta }$ is given in Equation (Equation 7).
(7)$$L\left\{D^{\beta }f\left(t\right)\right\}=s^{\beta }F\left(s\right)$$
when n−1<β≤n and $f\left(0\right)=f^{{}^{\prime }}\left(0\right)=\cdots =f^{\left(n-1\right)}\left(0\right)=0$.

It is important to note that the classical derivative of a function at an instant *t* is a local operator. However, the fractional derivative of a function at time *t* depends on past values, and it is therefore an operator with memory.

### 3.1. Fractional Systems

A non-integer linear time-invariant system with input *u*(*t*) and output *y*(*t*) can be represented in Equation (Equation 8).
(8)$$\sum_{k=0}^{n}a_{k}D^{\alpha_{k}}y\left(t\right)=\sum_{k=0}^{m}b_{k}D^{\beta_{k}}u\left(t\right)$$
where $\alpha_{k},\beta_{k}\in R$ and n≥m.

If the orders of derivation $\alpha_{k}$ and $\beta_{k}$ can be represented as a term *k*α, with *k* = 0, 1, 2,...the system is said to be of commensurate order Equation (Equation 9)
(9)$$\sum_{k=0}^{n}a_{k}D^{k\alpha }y\left(t\right)=\sum_{k=0}^{m}b_{k}D^{k\alpha }u\left(t\right)$$
and its transfer function is defined in Equation (Equation 10)
(10)$$G\left(s\right)=\frac{Y\left(s\right)}{U\left(s\right)}=\sum_{k=0}^{m}b_{k}{\left(s^{\alpha }\right)}^{k}/\sum_{k=0}^{n}a_{k}{\left(s^{\alpha }\right)}^{k}$$

It should be noted that a complex variable function such as Equation (Equation 11) is multi–valued. Its domain is a Riemann surface, with a finite number of sheets when $\forall k,\;\alpha_{k}\in Q^{+}$. The *q* sheets of the Riemann surface, with α = 1/*q*, are determined by
(11)$$\begin{array}{c} F\left(s\right)=\sum_{k=0}^{n}a_{k}s^{\alpha_{k}}\\ s=\;\left|s\right|e^{j\phi },\;\;\;\left(2k+1\right)\pi \;<\;\phi \;<\left(2k+3\right)\pi \; \end{array}$$
where k=−1,0,⋯,q−2. Note that only the roots of the principal sheet are meaningful [21].

The stability study of this type of control system is the key of its applicability. The stability analysis is performed, finding an integer index *m* such as $m\alpha_{k}$ which is an integer for k=0,1,…,n. Then, it is possible to define a transformation between the complex plane *s* and a new complex plane *v*, where $s=v^{m}$.

Figure 4 shows that the first Riemann sheet is a slice of the complex plane *v*, which is limited for a θ range of $\left(-\frac{\pi }{m},\frac{\pi }{m}\right)$. The line with $\theta =\frac{\pi }{2m}$ splits the first Riemann sheet into two zones. This line is the stability boundary and the zone above the stability boundary is the stability region [21,22,23].

### 3.2. $PI^{\alpha }D^{\beta }$ Fractional Controller

In the control theory, the classical PID has been modified by replacing the ordinary integral term for a fractional integral of order α, and by replacing the ordinary derivative term for a fractional derivative of order β. Indeed, Podlubny [24] proposed a generalization of the classical PID controller known as ${\mathrm{PI}}^{\alpha }D^{\beta }$, with 0 < α, β < 1. The fractional PID has two new tuning parameters (the fractional order of the integral and derivative actions) and it has shown a better performance in both time and frequency domains than its classical counterpart on some applications [25,26].

The ${\mathrm{PI}}^{\alpha }D^{\beta }$ controller expression in the time domain is shown in Equation (Equation 12) where e(t) is the error and u(t) the control input.
(12)$$u\left(t\right)=k_{p}e\left(t\right)+k_{i}D^{-\alpha }e\left(t\right)+k_{D}D^{\beta }e\left(t\right)$$

The transfer function of the ${\mathrm{PI}}^{\alpha }D^{\beta }$ controller is described in Equation (Equation 13).
(13)$$C\left(s\right)=K_{p}+\frac{K_{i}}{s^{\alpha }}+K_{d}s^{\beta }\;,\;\;\;\alpha ,\beta >0$$

## 4. Prototype Description

In order to carry out the controller design, a model of the traction response of the vehicle must be obtained first. For this, a constant voltage has been used as an open-loop input and the readings from the optical encoder coupled to the rear wheels are measured. This relates motor inputs with velocity output in open loop.

With the measured response, model adjustment has been made. Figure 5 shows the measured and model output for the same input. The right part of the figure corresponds to the part in which traction is not exerted and the vehicle stops due to the friction of the wheels with the ground.

The adjusted model cart is represented by the state variables described in Equation (Equation 14).
(14)$$A=\left[\begin{array}{cc} 0.00 & 1.00\\ -1.85 & -3.05 \end{array}\right];B=\left[\begin{array}{c} 0.00\\ 1.85 \end{array}\right];C=\left[\begin{array}{cc} 1.00 & 0.00 \end{array}\right];D=0.00$$

The prototype can also be described by the transfer function of Equation (Equation 15).
(15)$$G\left(s\right)=\frac{K}{\left(\tau_{1}s+1\right)\left(\tau_{2}s+1\right)}$$
with K=1, $\tau_{1}=1.20$, $\tau_{2}=0.45$.

## 5. Fractional Control Application

It should be highlighted that the introduction of fractional terms means that the dynamics of the closed-loop system does not depend on exponentials but on Mittag–Leffler functions described in Equation (Equation 16).
(16)$$E_{\alpha }\left(z\right)=\sum_{r=0}^{\infty }\frac{z^{r}}{\Gamma \left(1+\alpha r\right)}\;,\;\;\alpha >0\;and\;z\;\epsilon \;C\;$$
where Γ is the Gamma function. When α=1, the exponential is obtained as a particular case $E_{1}\left(z\right)=e^{z}$.

An important fact is that, unlike what happens with the product of two exponentials (Mittag–Leffler functions with α=1) which is another exponential function, the product of two Mittag–Leffler functions with α≠1 is not a Mittag–Leffler function but is obtained by Equation (Equation 17).
(17)EαaxEαay=∑r=0∞argαr; x, yΓ1 + αrgαr; x, y=∑i=0rriαxr−iyiriα=Γ1 + αrΓ1 + αiΓ1 + αr−i

It should be noted that, if α=1, this expression reduces to a binomial and the classical expression for the product of exponential is obtained. This effect has multiple consequences, but in this paper, the change in the time scale produced by the Mittag–Leffler functions is particularly interested. Thus, for rapid change signals, the dynamics are much faster than for an exponential, while for slow change signals, the opposite occurs, that is, the dynamics given by the Mittag–Leffler function is much slower than that of an exponential. To show this behavior in a simple way, the Mittag–Leffler functions for the simplest situation, represented in the fractional differential Equation (Equation 18), has been chosen.
(18)$$\frac{dy\left(t\right)}{dt}+c_{1}\frac{d^{\beta }y\left(t\right)}{{dt}^{\lambda }}+c_{2}I^{\alpha }y\left(t\right)=-y\left(t\right)$$

Note that the differential Equation (Equation 18) corresponds to a system with no input, and to observe the dynamics, an initial condition other than zero must be chosen. Thus, it was considered y(0)=2.

Figure 6a shows the behavior when the values $c_{1}=1;\;c_{2}=0$ were chosen. Only the dynamics generated by the fractional derivative term is present. Figure 6b shows the dynamic when $c_{1}=0;\;c_{2}=1$ have been chosen as parameters, so only the dynamics generated by the fractional integral term is present. The bandwidth of the controller can be adapted in function of the coefficients β in Figure 6a and α in Figure 6b, although the fractional controller gives more degrees of freedom to adjust the system behavior, changing the time response for the derivative and integral part.

The standard closed-loop transfer function of error versus reference is shown in Equation (Equation 19).
(19)$$\frac{E\left(s\right)}{R\left(s\right)}=\frac{1}{1+G\left(s\right)C\left(s\right)}$$

A controller $C\left(s\right)$ a ${\mathrm{PI}}^{\alpha }$ shown in Equation (Equation 20) is proposed.
(20)$$C\left(s\right)=\left(\frac{K_{p}s^{\alpha }+K_{i}}{s^{\alpha }}\right)$$
so the controller system transfer function is shown in Equation (Equation 21).
(21)$$\frac{E\left(s\right)}{R\left(s\right)}=\frac{s^{\alpha }\left(\tau_{1}s+1\right)\left(\tau_{2}s+1\right)}{s^{\alpha }\left(\left(\tau_{1}s+1\right)\left(\tau_{2}s+1\right)+KK_{p}\right)+KK_{i}}$$

The objective is to control a golf cart, so the possible commands that the path planning layer can send to the controller are a constant speed reference, and a speed change reference. Step (l=1) for constant speed and ramp (l=2) for change in the speed are considered as the possible inputs for the controller systems. The possible input references for the controller are shown in Equation (Equation 22).
(22)$$R\left(s\right)=\frac{r}{s^{l}}$$

As is well known, to calculate the stationary error, the final value theorem is applied in Equation (Equation 23).
(23)$$e_{stat}=\lim_{s\to 0}sE\left(s\right)\;=\lim_{s\to 0}\frac{rs^{\alpha +1}\left(\tau_{1}s+1\right)\left(\tau_{2}s+1\right)}{s^{\alpha +l}\left(\left(\tau_{1}s+1\right)\left(\tau_{2}s+1\right)+KK_{p}\right)+KK_{i}s^{l}}$$

If the reference is for the step type (l=1), the limit of Equation (Equation 23) is shown in Equation (Equation 24).
(24)$$e_{stat}=\lim_{s\to 0}\frac{rs^{\alpha }\left(\tau_{1}s+1\right)\left(\tau_{2}s+1\right)}{s^{\alpha }\left(\left(\tau_{1}s+1\right)\left(\tau_{2}s+1\right)+KK_{p}\right)+KK_{i}}\;$$
so, the final stationary error depends on α as shown in Equation (Equation 25)
(25)$$\left\{\begin{array}{cc} \alpha =0; & e_{stat}=\frac{r}{K\left(K_{p}+K_{i}\right)}\\ \alpha >0; & e_{stat}=0 \end{array}\right.$$

For this kind of reference, the classic PI can be used where α=1 and with zero error in the stationary. However, if the reference is ramp type (l=2), where the speed change from an initial value to a final one, the tracking stationary error can be calculated as Equation (Equation 26).
(26)$$e_{stat}=\lim_{s\to 0}\frac{r\left(\tau_{1}s+1\right)\left(\tau_{2}s+1\right)}{s\left(\left(\tau_{1}s+1\right)\left(\tau_{2}s+1\right)+KK_{p}\right)+KK_{i}s^{1-\alpha }}\;$$
the final stationary error depends on α as shown in Equation (Equation 27).
(27)$$\left\{\begin{array}{cc} c0\leq \alpha <1; & e_{stat}=\infty \\ \alpha =1; & e_{stat}=\frac{r}{KK_{i}}\\ \alpha >1; & e_{stat}=0 \end{array}\right.$$

In this case, the classical integer solution with α=2 obtains a zero stationary error, but it can make the closed-loop system unstable. For this reason, a fractional controller is used to achieve a zero stationary error, and it is necessary to carry out a stability analysis to assure stability. For this, it is considered as a final control transfer function Equation (Equation 28).
(28)$$G\left(s\right)C\left(s\right)=\frac{K}{\left(\tau_{1}s+1\right)\left(\tau_{2}s+1\right)}\left(\frac{K_{p}s^{\alpha }+K_{i}}{s^{\alpha }}\right)$$
and the frequency response must be calculated according to Equation (Equation 29).
(29)$$\begin{array}{c} mag=20log_{10}\left(\frac{\sqrt{K_{p}^{2}\omega^{2\alpha }+K_{i}^{2}+2K_{p}{K_{i}\omega }^{\alpha }cos\left(\frac{\pi }{2}\alpha \right)}}{\omega^{\alpha }}\right)\\ +\;20log_{10}\left(\frac{K}{\sqrt{{\left(\tau_{1}\omega \right)}^{2}+1}\sqrt{{\left(\tau_{2}\omega \right)}^{2}+1}}\right)\\ fase=arctag\left(\frac{K_{p}\omega^{\alpha }sin\left(\frac{\pi }{2}\alpha \right)}{K_{p}\omega^{\alpha }cos\left(\frac{\pi }{2}\alpha \right)+K_{i}}\right)-\frac{\pi }{2}\alpha -arctag\left(\tau_{1}\omega \right)-arctag\left(\tau_{2}\omega \right) \end{array}$$

To also guarantee stability and robustness, the hypotheses described in [21,27] will be used. Phase margin $\varphi_{m}$ has typically been used as a measure of stability and robustness. Thus, the phase margin $\varphi_{m}$ will be considered to define the desired nominal damping of the system. On the other hand, the crossover frequency $\omega_{cg}$ that fixes the desired nominal speed of the response of the system will also be used.

In order to calculate the gain crossover frequency $\omega_{cg}$, the equality defined in Equation (Equation 30) must be verified.
(30)$${\left.\left|C\left(j\omega \right)G\left(j\omega \right)\right|\left(K_{p},K_{i},\alpha \right)\right|}_{\omega =\omega_{cg}}=1$$

This value will depend on the parameters that characterize the controller, that is $K_{p},K_{i},\alpha$. At the frequency $\omega_{cg}$, the phase margin $\varphi_{m}$ is calculated according to Equation (Equation 31).
(31)$${\left.arg\left[C\left(j\omega \right)G\left(j\omega \right)\right]\left(K_{p},K_{i},\alpha \right)\right|}_{\omega =\omega_{cg}}=-\pi +\varphi_{m}$$

The two previous conditions by imposing values for $\omega_{cg}$ and $\varphi_{m}$ are established. Thus, the three parameters of the controller $K_{p},K_{i},\alpha$ are set as unknowns, a third condition that sets the phase of the open-loop system to be flat at $\omega_{cg}$ and consequently to be approximately constant in an interval around $\omega_{cg}$ according to Equation (Equation 32) is defined. The value obtained for α is fixed greater than 1, a condition which has been previously seen as necessary to achieve zero steady-state error when faced with ramp-type references.
(32)$${\frac{d\left(arg\left[C\left(j\omega \right)G\left(j\omega \right)\right]\right)}{d\omega }\left.\left(K_{p},K_{i},\alpha \right)\right|}_{\omega =\omega_{cg}}=0$$

The third condition establishes robustness against gain variations which guarantees robustness locally. The gain range depends on the frequency range at approximately $\omega_{cg}$ for which the phase keeps flat. This frequency range will be longer or shorter depending on the controller and the process.

## 6. Methods Discussion

The path-planning algorithm for the autonomous vehicle is based on a search in a space of the possible movements for the robot [28,29]. The path is divided in primitives; small actions can combine to make complex robot movements. The primitives of the cart include, different steering wheel angles and different displacement speeds. The combination of these primitives can compose any desired movement, and the path-planning algorithm joins the primitives looking for the best path.

The position of the steering wheel can be set accurately using a standard PID controller; however, a standard controller cannot accurately track the desired translation speed generated by the primitive. Focusing on cart movement primitives, 3 different primitives can be can highlighted.

The cart keeps the actual speed, which is equivalent to a step reference (l=1);The cart increases its speed, which is equivalent to a ramp reference (l=2);The cart reduces its speed, which is equivalent to a ramp with negative slope (l=2).

Constant speed can be kept by a standard controller with zero stationary error, so it can be assumed that this primitive is correctly followed. However, the primitives of increasing or decreasing speeds are different; this kind of command involves a ramp command, so the speed increases or decreases from one starting speed to a final one. These primitives are very difficult to follow by a standard controller, and the tracking error for this kind of command can be high. If the primitive is not followed correctly, the final cart control will be poor, and the cart performance can be limited.

The fractional control proposed in this paper is a practical solution for the cart speed control. This implementation improves the performance of the whole system, so the primitives are correctly followed, and the movement of the robot is similar to that planned by the path-planning algorithm.

## 7. Results

As mentioned, the design process consists of setting the values of the crossover frequency $\omega_{cg}$ and the phase margin $\varphi_{m}$. Figure 7 shows the results for two values of the crossover frequency $\omega_{cg}$, and the effect it produces on the Bode diagram. In both cases, it can be observed how for the value of the crossover frequency that $\omega_{cg}$ the phase reaches a maximum, and therefore, the derivative is zero. This fact corresponds to the robustness condition imposed. However, the overall width of the maximum in the phase diagram decreases as the crossover frequency $\omega_{cg}$ increases, and therefore, the overall robustness decreases.

Figure 8 presents temporal simulations that show how the error is reduced when the values of the $K_{p}$ and $K_{i}$ parameters are increased. Note that the vertical scale on which the error is represented changes. On the other hand, as the α value increases, the response becomes faster, but also more oscillating. Furthermore, for all values of $K_{p}$ and $K_{i}$, the closed-loop system becomes unstable when α=2, as shown in Figure 8d.

Table 1 shows a stability analysis for some representative cases presented in Figure 8.

In order to evaluate the proposed fractional PI controller, a series of experiments were conducted involving an electrical vehicle following different movement primitives. The goal of the vehicle was to maintain the desired speed with the smallest error. To facilitate this, the vehicle had to control its own power according to the path. The command can change a lot for the same speed, depending on the slope of the road, the pavement roughness, the battery level etc. The experiments were conducted using Simulink with the Real-Time Workshop toolbox, and the vehicle was is obtained from the odometer sensor. The set point for the Simulink model, which included the fractional controller, was the movement primitive generated by the path planning module, and the control action was transmitted to the vehicle’s control hardware via a serial protocol. The tests were carried out under different slope, pavement, and battery conditions. The objective of this paper was to improve the longitudinal controller for an autonomous vehicle. To measure the performance of the reference tracking, the difference between the reference velocity and the actual velocity of the prototype is used as a metric. If the reference tracking is good, the vehicle will be able to better follow the high-level primitives. This means that the maneuverability will increase, reducing the tracking error. High level layers will correct the control error introduced by system control, but if we reduce this error, the performance of the whole prototype will increase.

Figure 9 shows the cart speed error during two experiments following different primitives. From 0 to 10 s, the cart receives a movement primitive of acceleration, and should follow a speed ramp. The standard PID controller cannot follow the reference and it maintains a constant error, however, the fractional controller reduces the error over time. From 10 to 25 s, a constant speed primitive is set. The traditional integer controller significantly reduces the error, but the fractional controller reduces the error almost to 0. For the two real tests presented, the values $K_{p}$ and $K_{i}$ are maintained as fixed, while the value of alpha has been changed. The results obtained correspond to what was predicted by the simulations, where in Figure 9b, the error is reduced when α is increased and the system remains stable.

Figure 10 shows the ratio between the control command effort of a fractional strategy versus an integer strategy. The command is bigger for the fractional controller, and when α is increased, the ratio also grows. This is the expected behavior, so the error using a fractional controllers is also smaller.

## 8. Implementation of Fractional Module $s^{\alpha }$

The values that have been used for α as a fractional order coefficient in the real tests have been α=1.2 and α=1.4. In the actual implementation, two fractional modules have been used with α=0.5 and α=0.7. To obtain the value α=1.2, a 0.5 module and a 0.7 module were connected in series, while two 0.7 modules were connected in series to obtain the value α=1.4.

The Matsuda approximation has been used to obtain the two modules. First, a frequency range is chosen between a lower frequency $\omega_{l}$ and a higher frequency $\omega_{h}$ where the approximation is valid. It is also necessary to give the degree *n* of the approximation, which will determine N=2n. Then, *N* + 1 logarithmically distributed frequencies are calculated in the range of $\left[\omega_{l},\omega_{h}\right]$, Equation (Equation 33).
(33)$$\omega_{k}=\omega_{l}{\left(\frac{\omega_{h}}{\omega_{l}}\right)}^{\frac{k}{N}}\;k=0,\ldots ,N$$
and *N* + 1 coefficients are defined for each frequency $\omega_{k}$ which we will call $d_{i}\left(\omega_{k}\right)$ Equation (Equation 34).
(34)$$d_{i}\left(\omega_{k}\right)=\left\{\begin{array}{c} {\left(\omega_{k}\right)}^{\alpha }\;\;\;\;\;\;\;\;\;\;\;\;\;\;\;\;\;i=0;k=0,\ldots ,N\\ \frac{\omega_{k}-\omega_{i-1}}{d_{i-1}\left(\omega_{k}\right)-d_{i-1}\left(\omega_{i-1}\right)}\;\;\;i=1,\cdots ,N;k=0,\ldots ,N \end{array}\right.$$

Note that these coefficients must be calculated recursively. From the $d_{i}\left(\omega_{k}\right)$, we will define $c_{k}$ as Equation (Equation 35) shows.
(35)$$c_{k}=d_{k}\left(\omega_{k}\right)=\left\{\begin{array}{c} \omega_{0}^{\alpha }\;\;\;\;\;\;\;\;\;\;\;\;\;\;\;\;\;\;\;\;\;k=0\\ \frac{\omega_{k}-\omega_{k-1}}{d_{k-1}\left(\omega_{k}\right)-d_{k-1}\left(\omega_{k-1}\right)}\;\;\;\;k=1,\ldots ,N \end{array}\right.$$

With the $c_{k}$ values, it is possible to write the following truncated continued fraction expansion that approximates $s^{\alpha }$, as in Equation (Equation 36).
(36)$$s^{\alpha }≅c_{0}+\frac{s-\omega_{0}}{c_{1}+\frac{s-\omega_{1}}{c_{2}+\frac{s-\omega_{2}}{c_{3}+\frac{s-\omega_{3}}{c_{4}+\cdots }}}}$$

It is usual to write Equation (Equation 36) in a compact way by using the following notation, Equation (Equation 37).
(37)$$s^{\alpha }≅c_{0}+\frac{s-\omega_{0}}{c_{1}+}\frac{s-\omega_{1}}{c_{2}+}\cdots \frac{s-\omega_{N-1}}{c_{N}}$$

Note that since *N* is even, the degree of the numerator and denominator coincide. If *N* is odd, the degree of the numerator is one greater than the degree of the denominator. For this reason, N=2n was chosen. Table 2 shows the values of $c_{k}$ for the ninth-order approximation used for the modules with α=0.5 and with α=0.7.

Performing operations on the above equation can be easily reduced to a quotient of polynomials in *s* as Equation shown in (Equation 38).
(38)$$s^{\alpha }≅\frac{N\left(s\right)}{D\left(s\right)}=\frac{\sum_{J=0}^{n}b_{j}s^{j}}{\sum_{J=0}^{n}a_{j}s^{j}}$$

In this case, a ninth-order approximation, i.e., n=9 is chosen. Table 3 shows the values of $a_{k}$ and $b_{k}$ for the ninth-order approximation used for the modules with α=0.5 and with α=0.7. In Figure 11, the frequency representations of Matsuda approximations of different orders are shown, proving that the ninth order is a good approximation.

To obtain the discrete version, we used Tustin’s discretization, as in Equation (Equation 39).
(39)$$s≅\frac{2}{T}\left(\frac{1-z^{-1}}{1+z^{-1}}\right)$$
where $z^{-1}$ is the delay operator.

The approximation between the actual module in function of the *N* coefficient is shown in Figure 11, where both modules with α=0.5 and α=0.7 and its adjustment in function of the approximation degree *N* are shown. The ninth-order approximation follows in the frequency range the behavior of the fractional order controller with a negligible error. The computation cost of the implementation of these modules is also very small.

## 9. Discussion

As the results section shows, the use of fractional-order controllers represents a clear improvement in system control. When tracking control primitives for an autonomous vehicle, it is able to track them with less error than traditional controllers. Specifically, when the received command is a ramp, which is equivalent to a speed change in a certain slope, the fractional controller is capable of following it with an error in the stationary state of 0.

To achieve an equivalent performance using traditional non-fractional systems, it is necessary to use a double PID, however, this compromises the stability of the system. The use, as has been demonstrated in previous sections, of a PID with integral index α>1 allows obtaining a stationary error 0, but guarantees stability.

The tests carried out in simulation demonstrate that bandwidth and gain adjustment can be carried out with this type of controller. We also check how the index of the integral part of the PID affects the stability of the system, ensuring a stable value with correct tracking and a low stationary error with a coefficient of α=1.4. This demonstrates the better performance of this type of controller compared to the traditional ones.

The tests carried out on the real prototype confirm these results, with a much lower primitive tracking error than the previously used PID controller. The difference in computation time and complexity are clearly compensated thanks to the better performance of the overall system.

Using this type of controller, a more reliable autonomous vehicle system is obtained, capable of better following trajectories and performing more precise maneuvers, thus facilitates the control of high-level systems.

## 10. Conclusions

The low-level controller of an autonomous vehicle can make the difference in the performance of its activity. In this case, the analysis and implementation of the traction motor control for an autonomous cart is presented. A traditional PID control generates stationary output errors in the controller variable, but it is not valid for tracking speed changes, so a solution looking for a better tracking performance is presented.

In this article, a fractional ${\mathrm{PI}}^{\alpha }$ controller, with a parameter α>1 has been proposed for the speed ramp tracking problem of an electric car. It must be taken into account that the approach normally used in the literature considers fractional orders within the interval (0, 1].

Several simulations have been carried out that allowed demonstrating the better performance of the ${\mathrm{PI}}^{\alpha }$ controller, as well as an implementation in the electric vehicle that showed a remarkable reduction in the error.

The controller was applied to an autonomous electric cart, improving the low-level control performance and obtaining a better path tracking. The ability to follow more closely follow the trajectory facilitates the high-level tasks. This controller facilitates navigation in narrow areas and with multiple obstacles. 

## Figures and Tables

**Figure 1 sensors-23-03191-f001:**
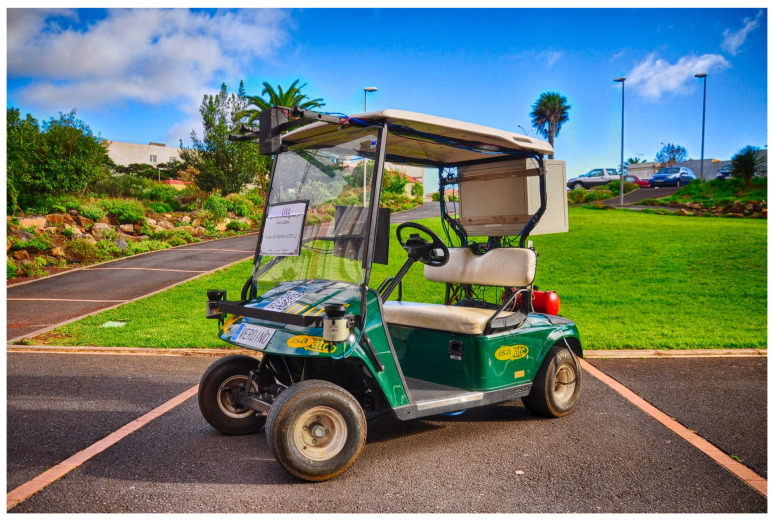
The sensors system to measure cart speed.

**Figure 2 sensors-23-03191-f002:**
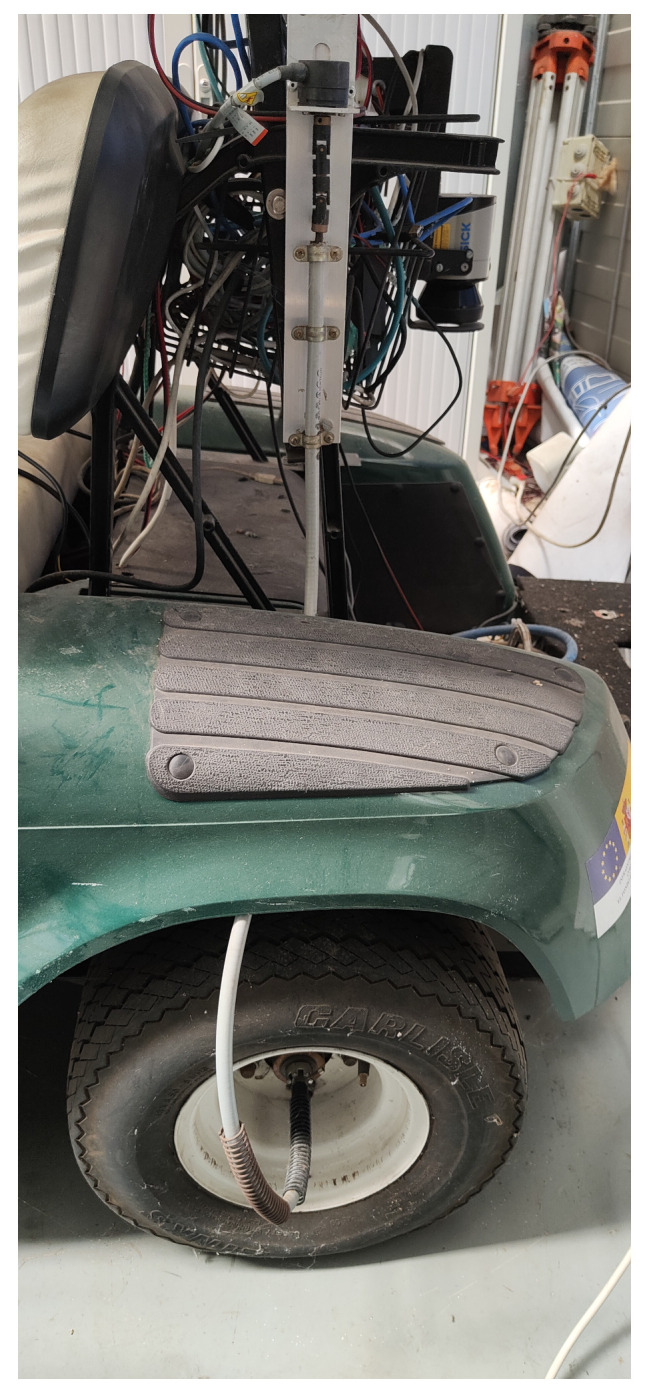
The odometric sensor coupled to the rear wheels.

**Figure 3 sensors-23-03191-f003:**
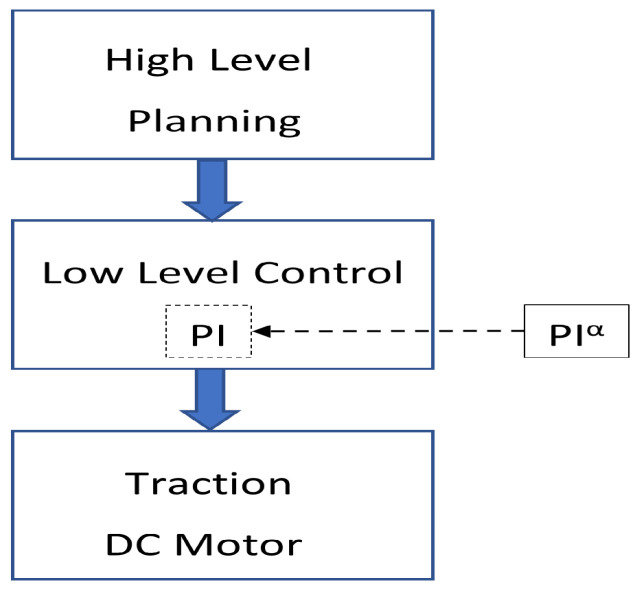
Overall block diagram of the autonomous cart.

**Figure 4 sensors-23-03191-f004:**
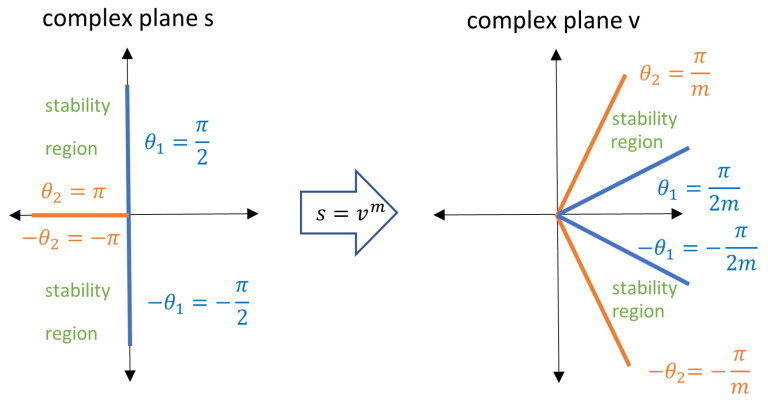
Transformation of the stability region from plane s to plane *v*.

**Figure 5 sensors-23-03191-f005:**
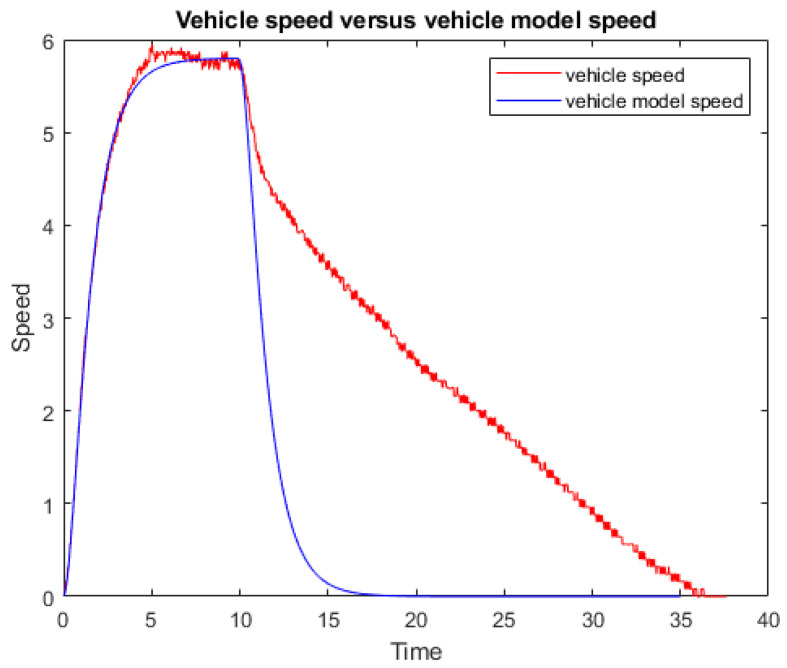
Step response for the electric vehicle.

**Figure 6 sensors-23-03191-f006:**
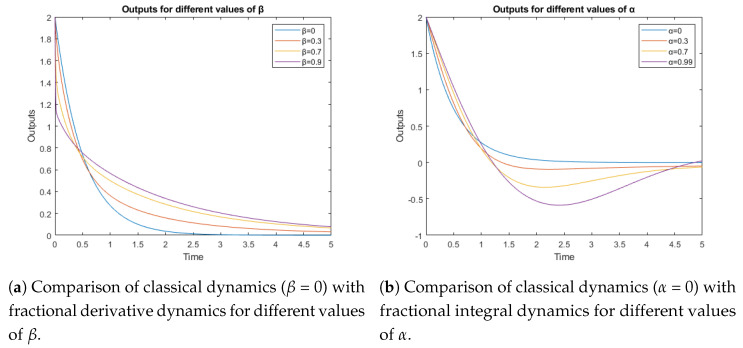
Behavior of the fractional terms with different parameters.

**Figure 7 sensors-23-03191-f007:**
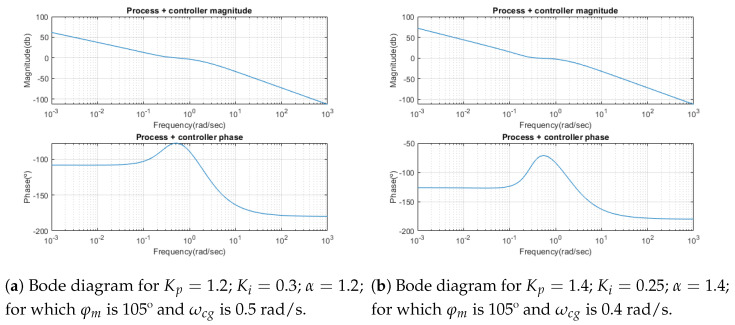
Bode plot of the system with different parameters.

**Figure 8 sensors-23-03191-f008:**
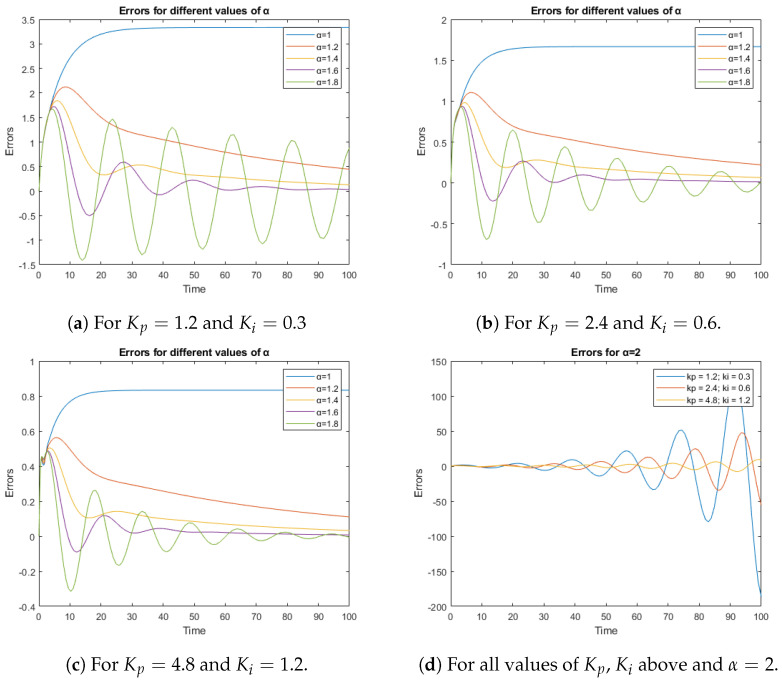
Tracking errors are shown for a ramp reference in different situations.

**Figure 9 sensors-23-03191-f009:**
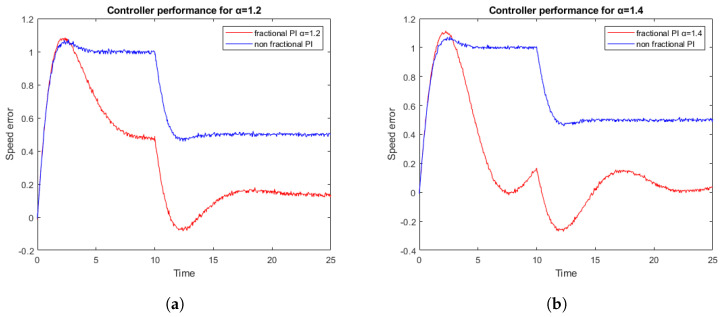
Tracking errors for a fixed reference of 2.5 m with $K_{p}=1.2$; $K_{i}=1$. (**a**) For a PI versus ${\mathrm{PI}}^{1.2}$; and (**b**) For a PI versus ${\mathrm{PI}}^{1.4}$.

**Figure 10 sensors-23-03191-f010:**
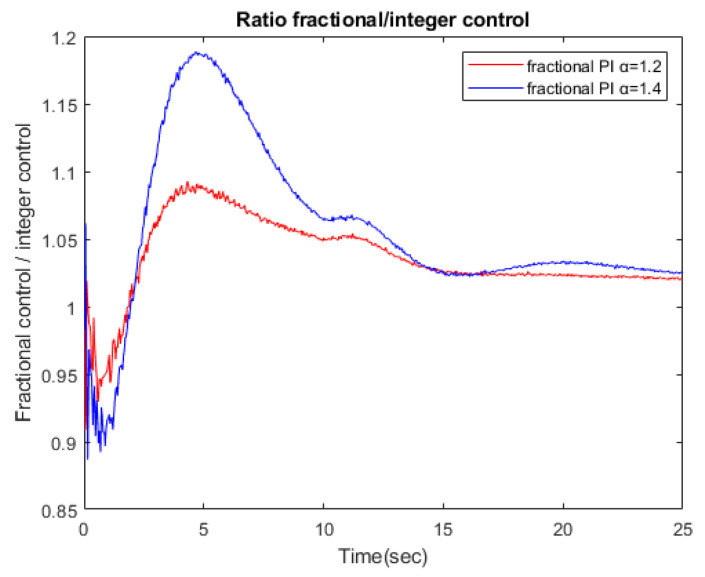
Ratio fractional control/integer control for ${\mathrm{PI}}^{1.2}$ versus ${\mathrm{PI}}^{1.4}$.

**Figure 11 sensors-23-03191-f011:**
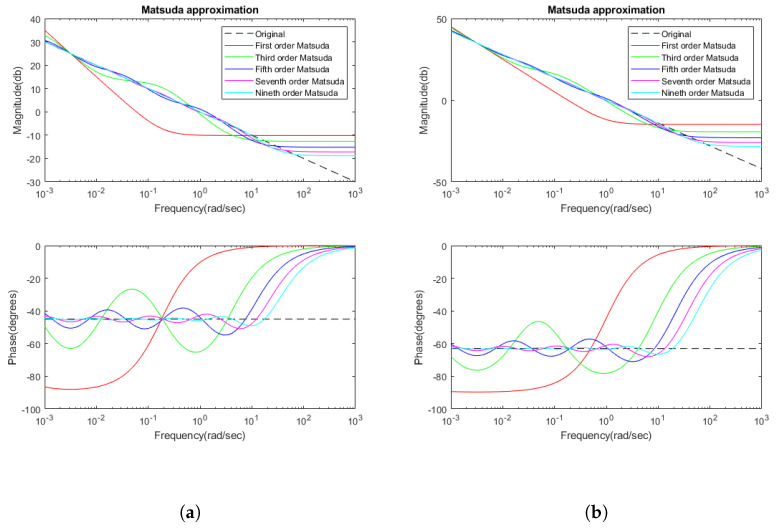
Magnitude and phase responses of Matsuda approximation of different orders: (**a**) for $s^{0.5}$; and (**b**) for $s^{0.7}$.

**Table 1 sensors-23-03191-t001:** Poles in the first sheet of the Riemann surface for some representative cases presented in the temporal simulations. The case of the last row corresponds to a situation where α=2.2, which is included to illustrate the presence of poles in the instability region.

Parameters	m	Poles in Stability Region	Poles in Instability Region
$K_{p}=1.2\;$ $K_{i}=0.3;$ α=1.2	5	1.0059 + 0.5396i 1.0059 − 0.5396i 0.6407 + 0.3570i 0.6407 − 0.3570i	———- ———- ———- ———-
$K_{p}=2.4$; $K_{i}=0.6$; α=1.4	5	1.0768 + 0.5192i 1.0768 − 0.5192i 0.7177 + 0.3305i 0.7177 − 0.3305i	———- ———- ———- ———-
$K_{p}=4.8$; $K_{i}=1.2$; α=1.8	5	1.1590 + 0.5089i 1.1590 − 0.5089i 0.7945 + 0.2773i 0.7945 − 0.2773i	———- ———- ———- ———-
$K_{p}=4.8$; $K_{i}=1.2$; α=2	1	−1.5566 + 2.8745i −1.5566 − 2.8745i	0.0302 + 0.4543i 0.0302 − 0.4543i
$K_{p}=1.2$; $K_{i}=0.3$; α=2.2	5	1.0213 + 0.5399i 1.0213 − 0.5399i	0.8001 + 0.2129i 0.8001 − 0.2129i

**Table 2 sensors-23-03191-t002:** Coefficients of the continued fraction expansion for the ninth-order Matsuda approximation.

A	Coefficients C
0.5	$C_{0}=10^{-3}\;;\;C_{1}=2.5647^{-3}\;;\;C_{2}=4.0132^{-3}\;$ $C_{3}=6.2796^{-3}\;;\;C_{4}=9.8260^{-3}\;;\;C_{5}=1.5375^{-2}$ $C_{6}=2.4058^{-2}\;;\;C_{7}=3.7645^{-2}\;;\;C_{8}=\;5.8905^{-2}$ $C_{9}=9.2172^{-2}\;;\;\;C_{10}=1.4423^{-1}\;;\;C_{11}=2.2568^{-1}$ $C_{12}=3.5313^{-1}\;;\;C_{13}=5.5256^{-1}\;;\;C_{14}=8.6461^{-1}$ $C_{15}=1.3529\;;\;C_{16}=2.1170\;;\;C_{17}=3.3125$ $C_{18}=5.1832$
0.7	$C_{0}=6.3096^{-5}\;;\;C_{1}=2.6337^{-2}\;;\;C_{2}=6.7040^{-4}$ $C_{3}=2.9510^{-2}\;;\;C_{4}=2.6694^{-3}\;;\;C_{5}=4.6540^{-2}$ $C_{6}=9.7623^{-3}\;;\;C_{7}=7.7435^{-2}\;;\;C_{8}=3.4763^{-2}$ $C_{9}=1.3109^{-1}\;;\;\;C_{10}=1.2257^{-1};\;C_{11}=2.2337^{-1}$ $C_{12}=4.3051^{-1}\;;\;C_{13}=3.8159^{-1}\;;\;C_{14}=1.5097$ $C_{15}=6.5256^{-1}\;;\;C_{16}=5.2909\;;\;C_{17}=1.1164$ $C_{18}=18.537$

**Table 3 sensors-23-03191-t003:** Coefficients of numerator and denominator of the ninth-order Matsuda approximation.

α	Numerator $N\left(s\right)$	Denominator $D\left(s\right)$
0.5	$b_{0}=8.76$ $b_{1}=52.260$ $b_{2}=30.508$ $b_{3}=2.4739$ $b_{4}=3.1015^{-2}$ $b_{5}=6.2017^{-5}$ $b_{6}=1.9589^{-8}$ $b_{7}=9.1993^{-13}$ $b_{8}=5.3200^{-18}$ $b_{9}=1.7783^{-24}$	$a_{0}=1$ $a_{1}=29.916$ $a_{2}=51.732$ $a_{3}=11.016$ $a_{4}=3.4874^{-1}$ $a_{5}=1.7441^{-3}$ $a_{6}=1.3912^{-6}$ $a_{7}=1.7156^{-1}$ $a_{8}=2.9388^{-15}$ $a_{9}=4.9261^{-21}$
0.7	$b_{0}=25.939$ $b_{1}=1.2228^{2}$ $b_{2}=58.519$ $b_{3}=3.9356$ $b_{4}=4.1069^{-2}$ $b_{5}=6.8328^{-5}$ $b_{6}=1.7874^{-8}$ $b_{7}=6.8520^{-13}$ $b_{8}=3.0830^{-18}$ $b_{9}=5.6234^{-25}$	$a_{0}=1$; $a_{1}=54.825$ $a_{2}=1.2185^{2}$ $a_{3}=31.784$ $a_{4}=1.2151$ $a_{5}=7.3032^{-3}$ $a_{6}=6.9986^{-6}$ $a_{7}=1.0406^{-9}$ $a_{8}=2.1745^{-14}$ $a_{9}=4.6127^{-20}$

## Data Availability

Not applicable.

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
