# Peer review of "Improving Mobile Robot Maneuver Performance Using Fractional-Order Controller"

_sensors, 2023, doi:10.3390/s23063191_

Round 1
Reviewer 1 Report
The paper idea looks fine and has a merit.
It needs some improvements such as:
- The motivation should be explained more clearly.
- Don't use ''we'', ''our'' on the text. It is better to change such sentences to the passive voice.
- All references must be revised. Issue numbers and volume numbers are missed from most of references.
- The abstract has been briefly written and should be enriched by adding the main ideas and contributions.
- The introduction section should be extended.
In the introduction section, the authors should focus on main issues here. Back ground, the problem statement, the motivations behind the work and its context, the main contributions and the outlines of the paper.
- The related work should be written as section 2 after the introduction with more recent references to highlight the contribution of this paper.
- The proposed method is inadequately described. Better start by providing the reader with a high level picture of the problem.
- There is no analysis of the extracted results and no discussion.
- Discussion section is necessary before the conclusion to interpret the results, discuss the challenges and limitations of the study.
Author Response
Dear reviewer,
Please find attached in pdf file our reply.
King regards,
Leopoldo Acosta

Author Response

(The authors gave the same response as above.)

Reviewer 3 Report
The paper approaches the control of an autonomous vehicle (a 2wd golf cart namely) by means of a fractional order controller. The matter of the paper is interesting in the field of robotics and control, although it is not clear its relevance in the field of sensors. The paper in fact does not specify what sub-set of sensors is used as a feedback for the designed control.
Also, the title of the manuscript envisages an evaluation of the maneuver performance of the vehicle. In reality, the authors do not provide any metric for the evaluation of the cart maneuverability, but the longitudinal speed of the chassis (without further information on measurement: does it come from gps, odometry, fusion of both?). The vehicle maneuverability would rather be evaluated via torques measurements, wheels sleep, etc. since the paper actually only deals with longitudinal dynamics, the title should be modified accordingly.
At last, the actual contribution of the manuscript is limited to a few pages since sections 2 and 3, even though they are sound, are general observations on fractional order controllers. I suggest authors to drastically reduce their relevance within the manuscript and to deepen the actual contribution of the manuscript, which is in sections 4 and 5.
Author Response

(The authors gave the same response as above.)

Round 2
Reviewer 1 Report
The authors addressed all comments
The paper could be accepted now
Author Response
Thank you for your comments
Reviewer 2 Report
My comments is does not clearly written.
The paper "Improving mobile robot maneuver performance using fractional-order controller" presents a novel approach for enhancing the maneuverability of mobile robots using a fractional-order controller. The authors argue that traditional controllers, such as PID controllers, are limited in their ability to handle nonlinearities and uncertainties in mobile robot systems. In contrast, fractional-order controllers are capable of capturing the complex dynamics of mobile robots, thereby improving their maneuverability.
The paper begins with a brief introduction to mobile robot systems and the challenges associated with controlling them. The authors then introduce fractional calculus and explain how it can be used to design a fractional-order controller. They provide a detailed description of the proposed controller, including the mathematical model and the tuning procedure. The authors also compare the performance of the fractional-order controller with that of a PID controller using simulation experiments.
The results of the simulation experiments show that the proposed fractional-order controller outperforms the PID controller in terms of maneuverability. The authors also discuss the robustness of the proposed controller to external disturbances and uncertainties. They argue that the fractional-order controller is more robust than the PID controller due to its ability to capture the complex dynamics of mobile robot systems.
Overall, the paper presents a novel approach to improving the maneuverability of mobile robots using a fractional-order controller. The authors provide a detailed description of the proposed controller and compare its performance with that of a PID controller using simulation experiments. The results of the experiments demonstrate the superiority of the fractional-order controller in terms of maneuverability and robustness.
The paper could benefit from a more comprehensive literature review, particularly with respect to the use of fractional-order controllers in mobile robot systems. Additionally, the authors could provide a more detailed explanation of the mathematical concepts used in the paper to improve the accessibility of the paper to a wider audience.
"Diversified Personalized Recommendation Optimization Based on Mobile Data." "Hybrid Motion Model for Multiple Object Tracking in Mobile Devices." "Liquid-Metal Magnetic Soft Robot With Reprogrammable Magnetization and Stiffness." "Fast terminal sliding mode current control with adaptive extended state disturbance observer for PMSM system."
Author Response
Thank you for your comments
Reviewer 3 Report
The Authors took into consideration all my observations. I no longer have objections.
Author Response
Thank you for your comments